# Potential Effects of Non-Surgical Periodontal Therapy on Periodontal Parameters, Inflammatory Markers, and Kidney Function Indicators in Chronic Kidney Disease Patients with Chronic Periodontitis

**DOI:** 10.3390/biomedicines10112752

**Published:** 2022-10-29

**Authors:** Ahmed Chaudhry, Nur Karyatee Kassim, Siti Lailatul Akmar Zainuddin, Haslina Taib, Hanim Afzan Ibrahim, Basaruddin Ahmad, Muhammad Hafiz Hanafi, Azreen Syazril Adnan

**Affiliations:** 1Periodontic Unit, School of Dental Sciences, Universiti Sains Malaysia, Kubang Kerian 16150, Malaysia; 2Hospital Universiti Sains Malaysia, Kubang Kerian 16150, Malaysia; 3Basic Sciences and Medical Unit, School of Dental Sciences, Universiti Sains Malaysia, Kubang Kerian 16150, Malaysia; 4Chemical Pathology Department, School of Medical Sciences, Universiti Sains Malaysia, Kubang Kerian 16150, Malaysia; 5Biostatics Unit, School of Dental Sciences, Universiti Sains Malaysia, Kubang Kerian 16150, Malaysia; 6Department of Neurosciences, School of Medical Sciences, Universiti Sains Malaysia, Kubang Kerian 16150, Malaysia; 7Advanced Medical & Dental Institute, Universiti Sains Malaysia, Bertam, Jalan Tun Hamdan Sheikh Tahir, Pulau Pinang 13200, Malaysia

**Keywords:** chronic kidney disease, inflammatory markers, periodontitis

## Abstract

Chronic kidney disease (CKD) and chronic periodontitis (CP) contribute to the increased level of inflammatory biomarkers in the blood. This study hypothesized that successful periodontal treatment would reduce the level of inflammatory biomarkers in CKD patients. This prospective study recruited two groups of CP patients: 33 pre-dialysis CKD patients and 33 non-CKD patients. All patients underwent non-surgical periodontal therapy (NSPT). Their blood samples and periodontal parameters were taken before and after six weeks of NSPT. The serum level of high-sensitivity C-reactive protein (hs-CRP), interleukin 6 (IL-6), and periodontal parameters were compared between groups. On the other hand, kidney function indicators such as serum urea and estimated glomerular filtration rate (eGFR) were only measured in CKD patients. Clinical periodontal parameters and inflammatory markers levels at baseline were significantly higher (*p* < 0.05) in the CKD group than in the non-CKD group and showed significant reduction (*p* < 0.05) after six weeks of NSPT. CKD patients demonstrated a greater periodontitis severity and higher inflammatory burden than non-CKD patients. Additionally, CKD patients with CP showed a good response to NSPT. Therefore, CKD patients’ periodontal health needs to be screened for early dental interventions and monitored accordingly.

## 1. Introduction

Chronic periodontitis (CP) is a chronic inflammatory disease caused by bacteria and affects the periodontium’s stability [1]. According to a report on the global economic impact of dental diseases, severe periodontitis is the sixth most prevalent disease globally, affecting 743 million people aged between 15 and 99 [2]. The development and maturation of plaque biofilm by bacterial colonization are considered the primary aetiological factor that contributes to periodontal disease’s pathogenesis [3]. The appalling consequences of periodontal disease include edentulism (tooth loss) and systemic inflammation [4].

Chronic kidney disease (CKD) is defined as reduced kidney function as indicated by glomerular filtration rate (GFR) having a value less than 60 mL/min/1.73 m^2^ and/or kidney damage for at least three months. Based on the degree of GFR reduction, CKD can be classified into five stages (stages I-V). The fifth stage (stage V) is considered an end-stage renal disease (ESRD), with a GFR value of less than 15 mL/min/l.73 m^2^ and persistent bilateral damage of the nephrons, which are the basic functional units of the kidney [5]. According to the World Health Organization (WHO) global health estimates, CKD is the world’s 14th leading cause of death [6]. In 2012, CKD caused 864,226 deaths (1.5% of deaths worldwide) [6]. CKD patients are susceptible to infection and atherosclerotic vascular disease, which are the major causes of morbidity and mortality in these patients. These two factors (i.e., infection and atherosclerotic vascular disease) were responsible for about 38% of annual deaths, with most of the deaths reported for ESRD patients [7]. A rise in inflammatory markers, such as interleukin 6 (IL-6) and C-reactive protein (CRP), are potent predictors of impaired kidney function and the development of cardiovascular disease in CKD patients [8,9].

Given the distinct pathogenesis of CKD and periodontitis, these two pathological conditions have been independent of each other. However, recent findings demonstrated a bidirectional relationship between CKD and periodontitis [10,11]. Moreover, clinical trials and cross-sectional studies suggested an association between CKD and the severity of periodontal problems [4,12,13]. CKD is responsible for a higher incidence of periodontal disease that is commonly manifested as plaque formation, calculus deposition, gingival hyperplasia, and increased gingival inflammation [14]. Furthermore, calculus and plaque build-ups have been associated with CKD patients’ uremic syndrome [15]. In other studies, CKD and CP have been associated with several risk factors, including impaired immunity, diabetes mellitus, smoking, impaired oral hygiene, xerostomia, and malnutrition. This may account for a link between CP and its deleterious systemic effects in CKD patients [16,17].

Non-surgical periodontal therapy (NSPT) is the keystone of periodontal therapy and the first recommended approach to preventing periodontal infections. NSTP is defined as “plaque removal, plaque control, supragingival and subgingival scaling root planning (SRP), and adjunctive use of chemical agents” [18]. Apart from its evident benefits on oral health or clinical periodontal aspects [19,20], periodontal therapy also improved endothelial function, reduced systolic and diastolic blood pressure, increased heat shock protein 10 (HSP-10; an anti-inflammatory factor), decreased white blood cells count, and reduced arterial intima-media thickness [21,22]. In addition, earlier investigations have shown that starting NSPT with local SRP is potent to reduce inflammatory markers in CP and CKD patients [23,24,25].

Based on the above evidence, we hypothesized that CKD patients have a higher prevalence of periodontal infections than non-CKD patients. Periodontal infections worsen the systemic inflammatory status, leading to poor renal outcomes in CKD patients. Improving oral health care through NSPT may improve the systemic inflammatory status and improve renal function. However, there are limited studies regarding the effect of NSPT on CKD patients. Therefore, this study aimed to investigate the changes in periodontal parameters, serum inflammatory markers, and kidney function indicators in CKD patients with CP and non-CKD patients following NSPT. The evaluation of such inflammatory markers may serve as important biomarkers for the diagnosis and monitoring of CKD. Targeted therapy to enhance these inflammatory markers may serve as a useful adjunct for treating CKD in its early stages and slow its progression to ESRD, which has an irreversible threatening effect on CKD patients’ morbidity and mortality.

## 2. Materials and Methods

### 2.1. Ethical Considerations

This study was submitted, reviewed, and approved by the Human Research Ethics Committee of Universiti Sains Malaysia (JEPeM USM code: USM/JEPeM/18020160). The study was conducted according to the guidelines described by the Helsinki Declaration of 1975, as revised in 2013. Informed consent was obtained from all patients prior to dental examination and the provision of periodontal therapy. All participants were aware of their right to withdraw from the study at any point during the study process.

A non-randomized clinical trial was conducted at Universiti Sains Malaysia (USM). A total of 66 subjects were divided equally into two groups. The non-CKD group comprised CP patients recruited from Dental Clinics Hospital USM, whereas the CKD group comprised CKD patients with CP recruited from the Nephrology Clinic and Chronic Kidney Disease Resource Centre, Hospital USM. The algorithm of patient recruitment in this study is presented in Figure 1.

### 2.2. Inclusion and Exclusion Criteria

The inclusion criteria for both groups were a) signing a written informed consent form, b) patients with moderate to severe periodontitis (i.e., clinical attachment loss (CAL) ≥ 1 mm and periodontal pocket depth (PPD) > 3 mm [26]), c) had at least 12 teeth in the oral cavity, and d) no scaling or root planning within the last six months. For the CKD group, only subjects in stage III and IV CKD determined based on the estimated glomerular filtration rate (eGFR) and had HbA1c levels < 7.5% were included [17].

The eGFR values were calculated to estimate the creatinine clearance using the Chronic Kidney Disease Epidemiology Collaboration (CKD-EPI) equation shown below:

eGFR = 141 × min (SCr/κ, 1) α × max (SCr/κ, 1) − 1.209 × 0.993 age × 1.018 [if female] × 1.159 [if Black] [27].

Exclusion criteria for both groups were (a) patients taking antibiotics, corticosteroids, statin, immunosuppressants, and aspirin which can affect the level of inflammatory markers in the past month, (b) pregnant women or lactating mothers, and (c) patients with a history of rheumatic fever, congenital heart disorders, prosthetic heart valves or any other condition that required them to have antibiotic prophylaxis prior to dental treatment.

### 2.3. Assessment of Clinical Periodontal Parameters

A proforma was used to collect patients’ demographic details, such as age, gender, ethnicity, and other comorbidities. A coding system was used for patient identification to ensure that only the researchers had access to their information. Only one examiner was assigned for clinical periodontal examination. Clinical data acquired by the clinical examiner were calibrated by senior specialists. The intra- and inter-examiner observations indicated about 90% of the recording being reproduced within a ±1.0 mm range. Briefly, the clinical periodontal examination was performed by assessing the CAL and PPD. The assessments were done by measuring six points for every tooth (except the third molar) using the Michigan probe with Williams marking.

### 2.4. Collection of Blood Samples

Five mL of blood samples were drawn from the median cubital vein and collected in a plain tube for the analyses of inflammatory markers (hs-CRP and IL-6) and kidney function indicators (serum urea and serum creatinine).

### 2.5. Provision of NSPT

Patients subsequently underwent NSPT, including SRP and oral hygiene instructions. Full mouth SRP using an ultrasonic (EMS Piezon Master, Electro-Medical System, Nyon, Switzerland), and curettage at PPD sites with 5 mm or greater using hand scalers (Gracey, Dentsply, UK) were performed under local anesthesia (Mepivacaine 2.2 ml with adrenaline ratio 1:100,000). All treated sites were then irrigated with 0.2% chlorhexidine. Patients were also given oral hygiene instructions to brush their teeth at least twice a day using fluoridated toothpaste and a soft-bristled toothbrush and floss once daily. Patients from both groups were followed up six weeks after the NSPT. Clinical periodontal parameters and blood samples were analyzed.

### 2.6. Biochemical Assay

The collected blood samples were centrifuged at 3000 rpm, and the blood serum was stored at −80 ℃ until they were analyzed. The serum hs-CRP level was measured using the latex particle-enhanced immunoturbidimetric method in a COBAS INTEGRA 400 plus (Roche Diagnostics, Basle, Switzerland). On the other hand, the IL-6 was measured using the electrochemiluminescence immunoassay method in the COBAS 6000 analyzer (Roche Diagnostics, Rotkreuz, Switzerland). Serum creatinine and urea levels were measured using a spectrophotometric method in the Architect C8000 analyzer (Abbott, KC, USA).

### 2.7. Statistical Analysis

The summary statistics were obtained using mean (SD) and frequency (percentage) for all variables. The independent *t*-test and Chi-squared test were used for comparing the CKD and non-CKD groups. Repeated-measures analysis of variance was used for comparing the periodontal parameters from the baseline to six weeks and between the groups. Analyses were carried out using IBM SPSS 24.0 Armonk, NY, USA with the significance level set at 5%.

## 3. Results

### 3.1. Primary Characteristics of Study Participants

A total of 87 subjects were initially recruited for this study. However, only 66 (43 males and 23 females) participated in this study (33 patients in each group). The mean age of patients in the non-CKD and CKD groups was 49.18 ± 8.58 years and 55.96 ± 11.26 years, respectively. In total, 89.3% (n = 59) of the patients were Malay, and only 10.6% (n = 7) were Chinese. In the CKD group, 12 and 21 patients were in stages Ⅲ and Ⅳ, respectively. Most of the patients were non-smokers, followed by ex-smokers and active smokers. The mean teeth count was significantly higher for the non-CKD group (*p* < 0.05) than for the (Table 1).

### 3.2. Clinical Periodontal Parameters

The PPD and CAL were significantly higher (*p* < 0.05) for the CKD group than the non-CKD group at the baseline level. Both groups showed significant improvement (*p* < 0.05) in the mean PPD and CAL after six weeks of NSPT. Nevertheless, there were no significant differences between groups when considering the mean of all periodontal variables during follow-up (*p* > 0.05). The mean baseline PPD for the non-CKD group was 4.76 ± 0.52 mm, while the mean PPD after NSPT was 2.97 ± 0.74 mm. On the other hand, the mean baseline PPD for the CKD group was 5.02 ± 0.50 mm, while the mean PPD after NSPT was 2.74 ± 0.50 mm. Apart from that, the mean CAL for the non-CKD group was 4.79 ± 0.52 mm and 3.20 ± 0.78 mm at baseline and post-treatment, respectively. In contrast, the mean CAL for the CKD group was 5.34 ± 1.06 mm and 3.26 ± 0.94 mm, respectively. Both groups demonstrated significant improvement in the plaque and gingival indicators after six weeks (Table 2).

### 3.3. Inflammatory Markers

The results for hs-CRP and IL-6 are tabulated in Table 3. The mean baseline for hs-CRP and IL-6 in the CKD group were 3.07 ± 2.37 mg/L and 4.11 ± 2.84 pg/mL, respectively. On the other hand, the mean baseline for hs-CRP and IL-6 in the non-CKD group were 1.71 ± 1.64 mg/L and 2.54 ± 1.09 pg/mL, respectively. The independent samples *t*-test showed that the mean baseline for hs-CRP and IL-6 were significantly higher (*p* < 0.05) in the CKD group than in the non-CKD group. Analyses revealed a significant reduction (*p* < 0.05) in inflammatory marker levels when both groups are considered. For the CKD group, post-treatment measurements for hs-CRP and IL-6 were 1.50 ± 1.38 mg/L and 2.93 ± 1.47 pg/mL, respectively. For the non-CKD group, post-treatment measurements for hs-CRP and IL-6 were 0.82 ± 0.71 mg/L and 1.89 ± 0.63 pg/mL, respectively. Apart from that, kidney function indicators (eGFR and serum urea) showed improvement during the study’s timeframe. However, these indicators did not show significant changes (*p* > 0.05) after NSPT.

## 4. Discussion

The NSPT was the first recommended approach to control periodontal infections. Notwithstanding its advancement over the years, NSPT remains the “gold standard” for which all treatment methods are compared. The main aim of NSPT is to restore gingival health by eliminating the factors responsible for gingival inflammation, such as endotoxins, plaque, and calculus in the oral cavity. This debridement procedure was carried out using hand instruments (i.e., curettes and scalers) and staged in different sessions [18]. Several researchers in the past have reported a decrease in gingival recession and probing depth after NSPT. They also indicated lesser bleeding during probing, lesser inflammation, and gingival redness [19,28,29]. Previous studies suggested that initial NSPT, comprising local SRP, is very potent in reducing inflammatory markers levels [30,31,32].

In this study, males were dominant in the CKD group since males are more susceptible to the disease than females [33]. This gender-specific finding of CKD progression and prevalence can be due to factors such as lifestyle, proteinuria, renal structure, body mass index, hypertension, sex hormones, and hyperglycemia [33,34,35]. Apart from that, we successfully ascertained the effects of NSPT in CKD subjects with CP and non-CKD subjects. Both groups showed major improvement in the clinical periodontal parameters and inflammatory markers after NSPT. Also, kidney function indicators have been shown to improve after NSPT. However, the observed difference was not significant. The results also showed an increased occurrence of tooth loss in patients suffering from CKD is associated with increased periodontal disease severity. Nevertheless, other factors, such as old age and concomitant medical problems such as diabetes, may exacerbate these problems [6,36].

Our study highlighted that CKD patients had a more severe form of periodontal disease than non-CKD patients. Overall, higher periodontal disease severity in CKD patients than in non-CKD patients can be associated with many risk factors, such as malnutrition, uremic syndrome, xerostomia, compromised immunity, and low oral health awareness [37,38]. In addition, the risk of diabetes mellitus was high in both groups. Previous research observed an independent link between diabetes and CP since diabetes is closely linked with reduced wound healing, increased monocyte response to dental plaque antigens, and impacted neutrophil chemotactic response. All these factors cause higher local tissue impairment [39]. Moreover, hyperglycemia may increase inflammation, oxidative stress, and apoptosis, contributing to increased periodontal destruction [40]. Additionally, diabetes has been known as one of the established primary aetiologies of CKD [6]. Hence, diabetes in the CKD subjects (Table 1) may also have caused additional damage to the periodontal tissues. Nguyen et al. (2017) suggested that the increased CAL status may be due to changes in salivary content (e.g., urea and calcium), thus contributing to the development of calculus in periodontal disease [4].

All periodontal aspects, such as PPD, CAL PS), and GBI, showed significant improvement in both groups after NSPT. The results were consistent with previous research that showed NSPT significantly improved CAL levels and reduced PPD in moderate or severe periodontitis [20,41]. NSPT is considered a gold standard for managing chronic periodontitis. Sanz et al. (2012) stated that numerous studies had reported its potency towards improving periodontal health through the mechanical debridement of subgingival plaque biofilms [20]. Clinical reports suggested that NSPT reduces the total number of gingival sites that bleed during probing, facilitating a transition of oral microbiota from gram-negative to gram-positive bacteria. Additionally, NSPT reduces the number of microorganisms, including black-pigmented species and spirochetes, with a concomitant increase in coccoid cells [42].

The findings from this study were similar to those reported by Artese et al. (2010), where periodontal therapy outcomes for CKD patients were analyzed [43]. The results of this study were also in agreement with other studies [44,45], which observed and recorded the differences in the clinical periodontal aspect before and after treatment for CKD patients with CP. Furthermore, advice on post-treatment oral healthcare given to patients in both groups may have played a role in improving oral health.

Inflammatory markers (IL-6 and CRP) were elevated in both CP and CKD conditions [36,37]. Our report highlighted a significant increase in baseline inflammatory marker levels for CKD patients than non-CKD patients. These observations may reflect the severity of systemic health problems. Pro-inflammatory cytokines (i.e., IL-6) accumulate within the body due to renal excretion failure caused by compromised renal function. These cytokines (IL-6) could also be attributed to the elevated production of CRP by hepatic cells [46]. Compromised immunity, atherosclerotic processes, cardiovascular disease, persistent infections, and gut microbiota dysbiosis have been reported as the factors responsible for the elevated inflammatory burden in CKD patients [47,48]. Such an inflammatory response is understood to be among the strongest predictors of diminished clinical outcomes for CKD patients [49].

As observed in this research, the significantly poor periodontal condition could also be related to the elevated production of IL-6 and hs-CRP in CP subjects. Several studies have documented CP as an infectious condition and a non-traditional risk factor for CKD due to high systemic inflammation loads. The high systemic inflammation loads arise from periodontal inflammation, locally generated inflammatory mediators, and acute phase reactants (IL-1, IL-6, tumor necrosis factor-alpha (TNF-α), and CRP) [10,37,50].

Accompanying the improvement in periodontal parameters after NSPT, both groups were observed to have significantly decreased serum inflammatory markers (*p* < 0.05).It was hypothesized that the control of local inflammation could result in a reduced systemic acute-phase response [51]. This may contribute to periodontal therapy’s anti-inflammatory effects, which lead to a lesser overall pathogen load in the oral cavity, thus reducing systemic and local inflammatory markers.

Findings on inflammatory markers were also consistent with other studies [45,52]. By contrast, a study in Japan suggested a non-significant reduction in hs-CRP and IL-6 levels after NSPT in patients with periodontitis only. Nevertheless, the researchers asserted that Japanese people have lower serum hs-CRP and IL-6 levels relative to other populations. Furthermore, the small sample size used in their study may have contributed to the conflicting result. Although there was no significant difference in both inflammatory markers, an improvement trend was observed in hs-CRP levels after NSPT [53]. Ide et al. (2004) reported a clinically significant elevation in serum IL-6 and TNF-α levels after NSPT [54]. Specific studies have highlighted the beneficial aspects of NSPT as transitory and asserted that the clinical inflammatory indicators typically increased 12 months following the treatment [55,56]. Apart from that, NSPT has been observed to improve serum urea and eGFR levels in CKD patients. However, these findings were not statistically significant, possibly due to the premorbid medical condition and the small sample size.

These findings highlighted that periodontal therapy might have delayed CKD progression. Usually, CKD is progressive and associated with a sustained decline in renal function, as reflected by the reduction in eGFR. Its progression depends on CKD causes, albuminuria levels, acute kidney injury, uncontrolled blood sugar levels, and blood pressure dysregulation [6]. Hence, post-treatment improvement of mean eGFR for subjects in the CKD group suggested that NSPT may have reduced CKD progression. Existing literature concerning the effects of NSPT on the status of kidney indicators remains controversial. Previous research has observed significant benefits of periodontal therapy towards the improvement of eGFR in CKD patients [30,43]. Nevertheless, Chambrone et al. (2013) carried out a systemic review focusing on the effect of periodontal therapy on eGFR. The review concluded that there is a lack of evidence to support the hypothesis that periodontal treatment has positive effects on eGFR, considering the limited number of studies and varying methodologies.

This study provides a useful approach to the future management of CKD patients, focusing on the importance of monitoring oral hygiene which has often been neglected. Periodontal therapy should be part of the treatment in retarding the progression of CKD patients in the future. More studies should be performed to further enhance our knowledge in this research area to support the study findings.

## 5. Conclusions

Pre-dialysis CKD patients demonstrated good clinical periodontal and inflammatory responses after NSPT. Hence, an understanding of periodontal health and its benefits for pre-dialysis CKD patients should be emphasized. Multi-centered research with a large sample size is required to evaluate periodontal treatment effects on periodontal parameters, eGFR status, and inflammatory markers.

## Figures and Tables

**Figure 1 biomedicines-10-02752-f001:**
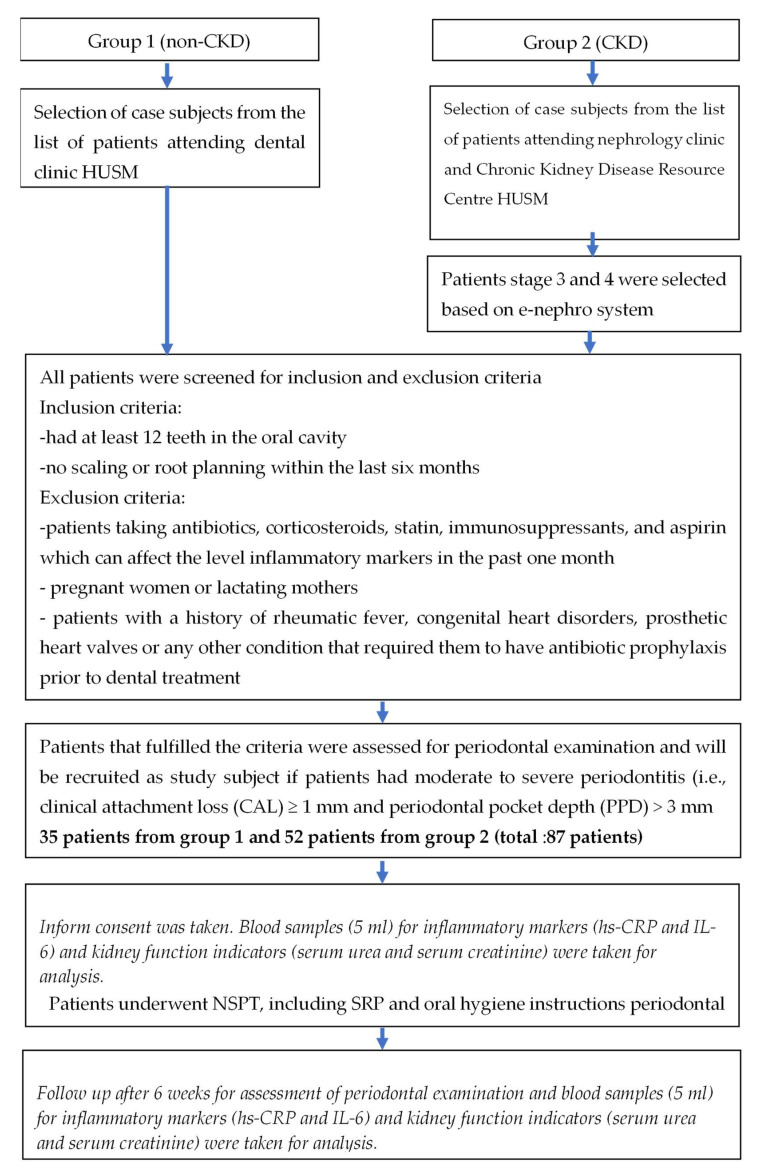
Algorithm of patient recruitment. HUSM, Hospital Universiti Sains Malaysia; CRP, C-reactive protein; IL-6, interleukin 6, NSPT, non-surgical periodontal therapy; SRP, scaling, and root planning.

**Table 1 biomedicines-10-02752-t001:** Demographic profile and medical status of the study population.

	Study Groups		Significance
Demographic Features	Non-CKD Group n (%) (n = 33)	CKD Group n (%) (n = 33)	*p*-Value
Age (years) (mean ± SD)	49.18 ± 8.58	55.96 ± 11.26	0.1
Gender			
Male Female	16 (24.2) 17 (25.8)	27 (40.9) 6 (9.1)	0.004
Ethnicity			
Malay Chinese	28 (42.4) 5 (7.6)	31 (47) 2 (3)	0.4
Marital status			
Married Single	28 (42.4) 5 (7.6)	27 (40.9) 6 (9.1)	0.7
Mean number of teeth present	24.36 ± 3.89	19.51 ± 6.23	< 0.001
Smoking			
Non-smokers Ex-smokers Active smokers	22 (33.3) 7 (10.6) 4 (6.1)	25 (37.9) 6 (9.1) 2 (3.0)	0.7
CKD staging			
Stage-III Stage-IV	- -	12 (36.4) 21 (63.6)	
Medical illness			
Diabetes mellitus Hypertension Ischemic heart disease Other medical illnesses No medical illness	1 9 - - 23	19 23 6 5 -	

Data are presented as mean (SD). *p*-values were determined using the chi-squared test. *p* < 0.05 was considered statistically significant. CKD, chronic kidney disease.

**Table 2 biomedicines-10-02752-t002:** Clinical periodontal parameters of the study population.

Periodontal Parameters	Baseline	Six Weeks Follow-Up	*p*-Value
PPD (mm)			*p*_time_ * _group_ = 0.002 *p*_time_ < 0.001 *p*_group_ = 0.9
Non-CKD	4.76 ± 0.52	2.97 ± 0.74 *	
CKD	5.02 ± 0.50 ^†^	2.74 ± 0.50 *	
*p*-value	0.046	0.15	
CAL (mm)			*p*_time_ * _group_ = 0.02 *p*_time_ < 0.001 *p*_group_ = 0.1
Non-CKD	4.79 ± 0.53	3.20 ± 0.152 *	
CKD	5.34 ± 1.06 ^†^	3.27 ± 0.152 *	
*p*-value	0.01	0.77	
GBI (%)			*p*_time_ * _group_ = 0.9 *p*_time_ < 0.001 *p*_group_ = 0.3
Non-CKD	52.78 ± 22.33	19.68 ±13.38 *	
CKD	56.12 ± 19.01	23.30 ± 5.81 *	
*p*-value	0.515	0.171	
PS (%)			*p*_time_ * _group_ = 0.6 *p*_time_ < 0.001 *p*_group_ = 0.3
Non-CKD	63.61 ± 18.41	23.49 ± 14.17 *	
CKD	61.53 ± 20.78	19.27 ± 8.16 *	
*p*-value	0.688	0.143	

Data are presented as mean (SD). *p*-values were determined using the independent *t*-test. *p* < 0.05 was considered statistically significant. * Statistically significant difference from baseline (*p* < 0.05).^†^ Statistically significant difference from the non-CKD group at baseline (*p* < 0.05). PPD, periodontal pocket depth; CAL, clinical attachment loss; PS, plaque scores; GBI, gingival bleeding index; CKD, chronic kidney disease.

**Table 3 biomedicines-10-02752-t003:** The value of inflammatory markers and kidney function indicator of the study population.

Variables	Baseline	Six Weeks Follow-Up	*p*-Value
hs-CRP (mg/L)			*p*_time_ * _group_ = 0.2 *p*_time_ < 0.001 *p*_group_ = 0.02
Non-CKD	1.71 ± 1.64	0.82 ± 0.71 *	
CKD	3.07 ± 2.37 ^†^	1.50 ± 1.38 *	
*p*-value	0.03	0.041	
IL-6 (pg/mL)			*p*_time_ * _group_ = 0.3 *p*_time_ = 0.001 *p*_group_ = 0.002
Non-CKD	2.54 ± 1.09	1.89 ± 0.63 *	
CKD	4.11 ± 2.84 ^†^	2.93 ± 1.47 *	
*p*-value	0.013	0.002	
Serum urea (mmol/L)			*p*_time_ * _group_ = 0.9 *p*_time_ = 0.8 *p*_group_ < 0.001
Non-CKD	4.15 ± 1.23	4.04 ± 1.15	
CKD	12.57 ± 4.84 ^†^	12.30 ± 5.35	
*p*-value	< 0.001	< 0.001	
eGFR (mL/min/1.73 m^2^)			*p*_time_ * _group_ = 0.6 *p*_time_ = 0.1 *p*_group_ < 0.001
Non-CKD	89.60 ± 21.33	92.21 ± 18.30	
CKD	25.96 ± 10.56 ^†^	27.18 ± 12.17	
*p*-value	< 0.001	< 0.001	

Data are presented as mean (SD). *p*-values were determined using the independent *t*-test. *p* < 0.05 was considered statistically significant. * Statistically significant difference from baseline (*p* < 0.05). ^†^ Statistically significant difference from the non-CKD group at baseline (*p* < 0.05). hs-CRP, high sensitive-C reactive protein; IL-6, interleukin-6; eGFR, estimated glomerular filtration rate; CKD, chronic kidney disease.

## Data Availability

Not applicable.

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
