# Peer review of "Potential Effects of Non-Surgical Periodontal Therapy on Periodontal Parameters, Inflammatory Markers, and Kidney Function Indicators in Chronic Kidney Disease Patients with Chronic Periodontitis"

_biomedicines, 2022, doi:10.3390/biomedicines10112752_

Round 1
Reviewer 1 Report
This study investigated the effect of non-surgical periodontal therapy (NSPT) on periodontal parameters, inflammatory markers, and kidney function indicators in chronic kidney disease (CKD) patients with chronic periodontitis. Although the findings is interesting, I haver several concerns about this study.
1. The study site and the method of patients enrollment were not mentioned. In addition, a figure to show the alogrithm of patients enrollment is needed.
2. Please describe the method of NSPT in the method section.
3. Please add some possible confounding factors, such as statin, corticosteroid which could affect the inflammation markes.
Author Response
Dear reviewer,
Thank you for your valuable comments. We have revised the manuscript accordingly. Please see the attachment.
Thank you

Reviewer 2 Report
In general, this work is really interesting and well written. My comments aim to increase the scientific soundness and clarity of it.
Line 8 – please correct affiliations (add departments)
Line 84 – short description of methodology applied should follow the goals.
Line 128 – from which vessel blood samples were collected ?
Line 140 – please provide country of origin for COBAS/Architect hardware
Line 147 – what was post-hoc test in ANOVA? What P-value was considered statistically significant?
Line 264 – please explain what TNF alfa stands for.
Author Response

(The authors gave the same response as above.)

Reviewer 3 Report
1. Glycaemic control with definition of HbA1c < 7% is very tight controlled, it is not "good" controlled. Nowadays, we do not suggest A1C <7% in CKD patients. Good is subjective, not objective.
2. Flowchart of inclusion and exclusion should be provided to get the total amount of patients that finally included.
3. How many patients had CKD stage 3A, stage 3B, and stage 4?
4. Serum urea is not correct term. BUN?
5. All abbreviations should be provided below all figures and tables
6. What are implications and future directions, please add in the discussion.
Author Response

(The authors gave the same response as above.)

Round 2
Reviewer 1 Report
The authors response well, so I have no more suggestion.